# A Constrained Programming Model for the Optimization of Industrial-Scale Scheduling Problems in the Shipbuilding Industry

Javier Pernas-Álvarez *[ID] and Diego Crespo-Pereira

Universidade da Coruña, Campus Industrial de Ferrol, CITENI, Grupo Integrado de Ingeniería, Campus de Esteiro s/n, 15403 Ferrol, Spain; diego.crespo@udc.es
* Correspondence: javier.pernas2@udc.es

**Abstract:** This work presents an innovative constrained programming model for solving a flexible job-shop scheduling problem with assemblies and limited buffer capacity based on a real case from the shipbuilding industry. Unlike the existing literature, this problem incorporates the manufacturing and assembly of blocks from subblocks to the final ship erection, while considering the limited buffer capacity due to the size of blocks, which has been often overlooked. The objectives considered are the minimization of the makespan and tardiness based on ship erection due dates. To demonstrate the model's effectiveness, it is initially validated using various scheduling problems from the literature. Then, the model is applied to progressively challenging instances of the shipbuilding problem presented in this work. Finally, the optimization results are validated and analyzed using a comprehensive simulation model. Overall, this work contributes to reducing the gap between academia and industry by providing evidence of the convenience of the application of constrained programming models combined with simulation models on industrial-size scheduling problems within reasonable computational time. Moreover, the paper emphasizes originality by addressing unexplored aspects of shipbuilding scheduling problems and highlights potential future research, providing a robust foundation for further advancements in the field.

**Keywords:** shipbuilding; MILP; CP; scheduling; optimization

## 1. Introduction

Shipbuilding is an extraordinarily complex and Engineering-to-Order industry where each order (ship) is managed as a customized project and involves an endless amount of resources and technologies [1–5]. Thus, each project entails a high degree of uncertainty and associated risk, leading to the need for methods and systems to plan, monitor, and control the production systems involved. Furthermore, the current global political context and derived conflicts have prompted a race toward digitalization and efficiency within the industry. Manufacturers are striving to reduce costs and lead times to become more competitive while keeping quality standards. In doing so, they have been adopting techniques and production methodologies that come traditionally from other industries such as Lean Manufacturing and Product Lifecycle Management (PLCM) [3,4,6].

Since ship manufacturing is large-scale and greatly non-standardized [7], the production process of a ship involves thousands of operations that are interrelated and many times performed in parallel, thus depending on each other. This is especially noticeable in the block assembly process, the problem that this paper addresses, which requires a high degree of coordination between resources to meet deadlines and avoid cost overrunning [7,8].

Currently, the construction of ships is mostly based on the assembly of blocks made in turn of subblocks which are assembled in cells [7–9]. Conversely, the scheduling of subblocks determines the availability of blocks, which in turn constrains the assembly strategy of the ship at the dock. On top of this, the characteristics and size of subblocks

and blocks make intermediate storage a critical element in operation scheduling. The timing of each operation must also be carefully considered to avoid costs associated with either the unavailability of sufficient storage space or the need to rent additional storage capacity. Thus, the construction sequence adopted in these stages not only influences the total completion time but also dictates the storage requirements, ultimately impacting the overall efficiency of the ship construction process. Based on this, following the notation of [10], a flexible job-shop scheduling problem with assembly operations and limited buffer capacity (FJSP-A-LBC) can be defined. Different computerized optimization techniques can be used to address this problem.

Refs. [1,10] point out that so far, few works have addressed the FJSP in shipbuilding. Mixed-Integer Linear Programming (MILP) as described in [5,7,10] and discrete-event simulation models like the ones used in [8,10–13] are the main approaches used in this area. Beyond shipbuilding, MILP is by far the most common approach to address the FJSP [14], although it has been frequently combined with other techniques. MILP-based hybrid approaches include heuristics [15–17], metaheuristics [15,18], and constrained programming (CP) [14]. Techniques of decomposition–aggregation and improvement algorithms like in [10] must be mentioned too. These alternatives to exact optimization methods provide reasonable computational times for large-sized cases at the expense of optimal solutions.

Since MILP models usually entail long computational times [18] in medium- and large-sized problems, another optimization approach that has recently been emerging as a serious alternative for scheduling problems is CP [14]. CP is an optimization approach to solving constraint satisfaction problems (CSP) [19,20] that has not yet received much attention from practitioners. This is due to several reasons such as semantics (CP is based on restrictions that are not as familiar as pure mathematical formulations), a certain skepticism of whether CP optimizers outperform other approaches on scheduling problems [19,21], and even commercial pressures [22]. The first efforts to incorporate CP into scheduling problems are based on Logic-based Benders Decomposition (LBBD) [19,22–24], a hybrid approach that combines CP and MILP. However, within CP, Constrained Integer Programming (CIP) has arisen as a promising optimization approach that seems to outperform both MILP and hybrid approaches like LBBD. This is stated in [22], where CIP models are able to solve more problems to optimality than Mixed Integer Programming (MIP) and LBBD models. In [24], the CP Optimizer (CPO), IBM's proprietary CP solver, is upgraded with interval and sequence variables [25], thus substantially reducing the number of variables. CPO outperforms the rest of the approaches in the instances examined in [24]. Ref. [25] recommends using CPO in industrial-size scheduling applications. For readers who are unfamiliar with CP, Refs. [20,26] provide a good overview of the history of CP and related software.

Ref. [26] also provides an explanation of CPO Automatic Search, CP's optimization algorithm. Different techniques are combined like constraint propagation [27], CP search tree, Large Neighborhood Search (LNS) [28], linear relaxation, failure-directed search (FDS) [29], and iterative diving along with parallelization. The criteria for the use of each of them depends on the size of the problem and the evolution of the optimization. For instance, if the problem is small enough or the solution is not being improved, FDS performs a complete search. Likewise, LNS performs a meta-heuristic search in medium- to large-sized cases. CP resorts to aggressive dives in the CP search tree from the iterative diving algorithm when the problem is too large for LNS. Ref. [20] provides a recent comparison between CPO and Google's OR-Tools (ORT) for the job-shop scheduling problem (JSP), concluding that CPO outperforms ORT in large-scale cases (limitations of the benchmarking must be considered). In fact, Ref. [20] expects a rise in the number of industrial applications of CP.

When it comes to the incorporation of limited buffers in job-shop scheduling problems, to the best of our knowledge, this aspect has received limited attention in previous research [30–32]; most efforts have been dedicated to the flow-shop scheduling problem. Moreover, it has been even less explored in the context of shipbuilding production sys-

tems. Ref. [33] already showed that the two-machine flow-shop problem with a limited buffer capacity between the first two machines is NP-hard. Most recent studies in the field like [14,19,20] consider either flow- or job-shop problems with assemblies but assume no constraint regarding buffer capacity, the latter thus becoming infinite, as in classical problems [32,34]. If we resort to other areas, buffering constraints have barely been included in the MILP model. Ref. [31] provides a good reference to classify job-shop problems according to the type and capacity of buffers:

- Output buffers: Machines have an output buffer downstream with limited capacity where the job can be stored once the operation on the machine is finished.
- Input buffers: Machines have an input buffer upstream with limited capacity where the job can be stored once the operation on the previous machine is finished.
- Pairwise buffers: Each pair of consecutive machines has a specific buffer to store the job, if necessary, when it goes from the machine upstream to the machine downstream.
- Job-dependent buffers: There is a dedicated buffer for each job, so the assignment of the operations of a job to buffers depends on the job itself.
- Blocking scheduling problem: A special case where buffers have no capacity, so operations may block machines if subsequent machines are busy.

Taking this notation as reference, Ref. [30] studies the multi-route job-shop scheduling problem with limited output buffers comparing a hybrid artificial immune-simulated annealing algorithm with a MILP model. Ref. [35] uses MILP to solve a blocking flow-shop model with up to 20 jobs and seven machines, which is still far from large-scale problems derived from shipbuilding. Ref. [36] investigates the job-shop problem of a robotic manufacturing cell with intermediate buffers. In their case, they consider restrictions on the time a manufactured piece can block a station if the downstream buffer is blocked: no-wait, free pick-up (unlimited time), and a time window (limited time). Ref. [37] proposes a MILP model to study a cyclic hybrid flow-shop problem with limited output buffer capacity, obtaining an assignment heuristic algorithm to generate initial sequences for the MILP model.

Beyond MILP, metaheuristics are the most common approach to address scheduling problems with buffering constraints. Ref. [32] applies a novel heuristic algorithm based on simulated annealing to the job-shop scheduling problem considering four different buffering constraints: no-wait, no-buffer, limited-buffer, and infinite-buffer. Ref. [38] uses tabu search to obtain good solutions for a flow-shop problem with limited buffer capacity. Ref. [39] applies an extended version of a genetic algorithm to optimize the makespan of a flow-shop problem with sequence-dependent setup times and output buffers with limited capacity.

For a comprehensive literature review, Ref. [40] provide insights on job-shop scheduling problems (JSP) and flow-shop scheduling problems (FSP) with buffering constraints. However, there is no study that specifically examines the job-shop scheduling problem with assembly operations and buffering constraints. Similarly, Refs. [31,32] offer references on flow-shop scheduling problems with buffering constraints, but do not address the specific combination of assembly operations and buffering constraints in the job-shop scheduling context.

Therefore, given the potential that CP seems to have in scheduling problems and the existing gap in the shipbuilding literature, this study first formulates a CP model of the FJSP-A for the case studies examined in [10] and compares the results of the minimization of the makespan between models. Since CP outperforms both the monolithic MILP formulation and the MILP-based decomposition algorithm proposed by [10] for the larger cases, a new variant of the FJSP with assemblies and limited buffer capacity is formulated and investigated. The problem derives from a real case from the shipbuilding industry and tackles the criticality of intermediate storage between stages due to the size of blocks and subblocks. Hence, several instances are defined based on buffer capacity, optimization objective, and number of blocks. MILP and CP models are formulated for each instance and a detailed comparison of the computational performance and the quality of the solutions is

presented. Discrete-event simulation models are used to further validate the results and obtain insights into various key performance indexes that cannot be directly extracted from the optimization. Overall, our primary objective is to bridge the gap between academic research and industrial practice by demonstrating the effectiveness of constrained programming on large-scale scheduling problems, particularly in shipbuilding, where efficient production plans and storage capacity limits are crucial. We strive to develop a computerized optimization methodology that can accommodate manufacturing complexities and provide efficient production plans within reasonable computational time. Additionally, the approach must be designed so that results can be easily communicated to non-expert personnel, thereby supporting decision-making processes in various stages of the project.

## 2. The Shipbuilding Manufacturing Process

The assembly process of subblocks and blocks is a complex production process that starts with the manufacturing of sheets and profiles, components of blocks. Following a high-level modeling approach, activities can be grouped into parent activities according to different criteria such as activity location, nature, or personnel involved. We grouped the activities according to the workshops and flows between them (Figure 1). In doing so, we consider the following workshops (W), each one containing several multipurpose cells:

- WA1, WA2: Workshops for assembly operations. Cells in WA1 and WA2 can also execute outfitting operations if needed.
- WO1, WO2: Workshops designed for outfitting operations. Cells belonging to WO1 and WO2 can also perform assembly operations if needed.
- BTC: Outside block turning cells for subblock turning.
- PC: Painting cabin.
- SW: Slipway for block erection.

We consider the following operations:

1. Subblock assembly 1 (SB-A1): Sheets and profiles that have been cut, formed, and welded are delivered to subblock assembly 1 area (SB-A1) to form subblock subassemblies. It can be executed in WA1 and WA2 cells.
2. Subblock assembly 2 (SB-A2): Subblock subassemblies are welded together to form subblocks. This operation can be continued in WA1 and WA2 workshops or executed in WO1 and WO2 if necessary.
3. Turning (SB-T): Some subblocks must be turned upside-down to proceed to block assembly. Turning can only be performed in BTC.
4. Block assembly (B-A): Subblocks are welded together to form blocks, which are the component parts of the frigate hull. This can be executed in WA1, WA2, WO1, WO2, or in the previous cell of BTC.
5. Outfitting 1 (B-O1): Piping, brackets, and other equipment fabricated in auxiliary workshops are installed on the block. This can be performed in WO1 and WO2 or in WA1 and WA2 if necessary.
6. Blasting and Painting (B-P): Block blasting and painting are performed in painting cabins. Blasting and painting can only be performed in PCs.
7. Outfitting 2 (B-O2): Electrical equipment, ducts, and other equipment fabricated in auxiliary workshops that could have been affected by the painting process are installed on the painted block. This can only be performed in WO2.
8. Block Erection: Once blocks are completed, they are erected and welded in a slipway to form the frigate hull according to a predefined strategy. Each block enables adjacent blocks to be erected and welded, so the production of blocks should be adjusted to hull construction to avoid unnecessary blockages and storage. This is executed in the SP.

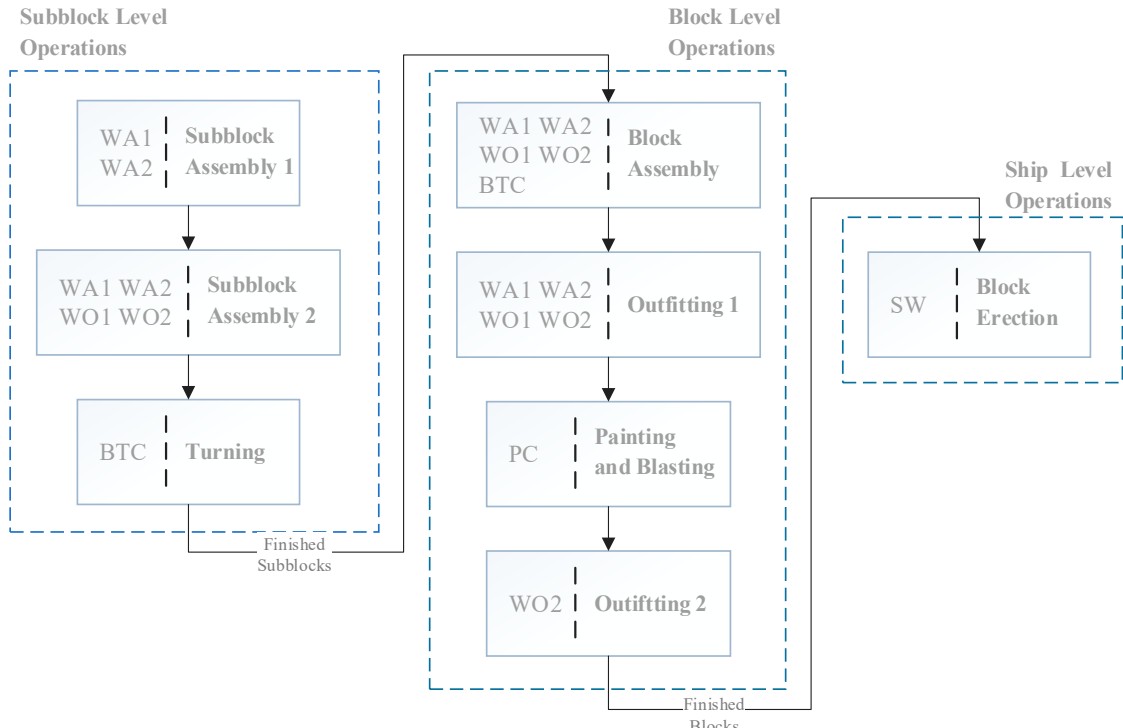

**Figure 1.** Process flow of the shipbuilding process.

Refs. [8,9] depict a similar process flow, but two extra operations (SB-A2 and SB-T) are included in this paper to consider the flow of parts between workshops. It is also remarkable how cells in different workshops are multipurpose and can hold different operations.

Finally, it is worth noting the impact of the block erection strategy on the scheduling problem. In this sense, we have considered a twofold objective: the makespan (MK) and the minimization of total tardiness according to predefined blocks' due dates (DD). The latter allows for the adjustment of the solutions to the scheduling problem to a predefined block erection strategy.

## 3. Materials and Methods

### 3.1. Problem Statement

The flexible job-shop scheduling problem with assembly operations considers the following assumptions:

- The set of jobs $J = \{j_1, j_2, \ldots j_n\}$ are to be processed in a set of stages $S = \{s_1, s_2, \ldots s_n\}$.
- Each job $j \in J$ is composed of a variable number of operations $o \in O_j$ to be executed in the set of stages $S$. For each job, there is a predefined sequence to execute the operations which depends on the type of job, not necessarily using all the stages.
- Only one operation of a given job can be processed at a time in a given stage.
- Each stage $s \in S$ is made up of a subset of workstations $w \in W_s$. A workstation can perform operations in various stages, thus belonging simultaneously to the given stages.
- One given workstation can only perform one operation at a time. No other equipment constraint is considered.
- For assembly operations, the set of jobs $J$ is split into two subsets $J^{sa}$ and $J^a$ to consider subassemblies $J^{sa}$ and final assemblies $J^a$.
- $O^a$ represents the set of the last operation of jobs $j \in J^a$
- Jobs $j \in J^{sa}$ are available to be scheduled at time 0.
- Jobs $j \in J^a$ can only start processing when all their subassemblies (complementary jobs $j' \in J^{sa}$) are completed.
- Operations of jobs $j \in J^{sa}$ follow always the same sequence of stages.

- Operations of jobs $j' \in J^a$ follow always the same sequence of stages which is different from jobs $j \in J^{sa}$ operations' sequence.
- Both jobs $j \in J^{sa}$ and $j' \in J^a$ may have operations assigned to the same stage $s \in S$.
- For a given operation $o \in O_j$ of a job $j \in J$, the processing time $PT_o^j$ is always known in advance.
- Transportation times of jobs between stages are negligible and thus ignored. The same happens with setup and changeover times.
- Machine breakdowns and preventive operations are not considered.
- All model parameters such as processing times are deterministic.

Two goals have been considered:

- The minimization of the makespan (MK), which is the total time to complete all the operations belonging to all jobs.
- The minimization of the total tardiness (DD), which is the sum of the tardiness of all jobs according to predefined due dates.

For the modeling of storage areas, we have taken a similar approach to [31]. Therefore, buffers are considered additional workstations in the model and mandatory steps for a job when an operation is completed in a stage. Therefore, when considering the FJSP with assembly operations and limited buffer capacity, we add the following assumptions:

- There is a set of storage areas $B = \{\beta_1, \beta_2, \ldots \beta_n\}$ to store jobs when the next operation cannot be processed for workstation availability.
- Each storage area $\beta \in B$ is composed of a subset of buffers $b \in B_\beta$ of one unit of capacity.
- The size of the storage area $\beta$ is given by the length of the subset of buffers $b \in B_\beta$.
- The size of the storage area $\beta$ is a positive integer that ranges from 0 to $\infty$. A size of $\infty$ for all storage areas simplifies the problem to the FJSP with assembly operations.
- For a given job $j \in J$, the list of operations $O_j$ is modified so that between every two operations $o - 1_s$ *and* $o_{s'}$, $o \in O_j$, $(s, s') \in S$, a new operation $o\prime$ is inserted and assigned to storage area $\beta_s$.
- The set of single-unit buffers $b \in \beta_s$, $s \in S$ are modeled as workstations $w \in W$ with a processing time of 0 or higher.

Based on these assumptions, two variants of the base case are considered in this paper according to [31]:

- Blocking FJSP with assembly operations: The size of all storage areas $\beta$ is null or 0.
- FJSP with assembly operations with output stage-dependent buffers: Each stage $s$ has been assigned an output storage area $\beta_s$ with limited capacity ($< \infty$) to move jobs when their operation in the current stage has been finished.

Finally, a mapping of the previous assumptions and definitions is performed for the shipbuilding case considered in this paper. Therefore:

- Jobs $J^a$ represent the blocks and jobs $J^{sa}$ represent the subblocks.
- A workstation $w \in W$ represents a workshop slot or cell for performing operations in a given block or subblock.
- Stages $s \in S$ are groups of workstations according to the shipbuilding operations shown in Figure 1. Therefore, each stage $s \in S$ represents a shipbuilding process and not a workshop.
- Operations $o \in O$ are the shipbuilding processes represented in Figure 1 from subblock "Assembly 1" to "Outfitting 2".
- Block erection operation is indirectly considered by means of due dates. Thus, block erection start dates are parameters of the model that considers due dates.
- Subblocks $J^{sa}$ follow the sequence of operations represented in Figure 1 from "Assembly 1" to "Block Assembly". Blocks $J^a$ follow the sequence of operations from "Outfitting 1" to "Outfitting 2".

### 3.2. Mathematical Formulations

#### 3.2.1. MILP Model

The mathematical formulation of the MILP model for FJSP-A-LBC is based on [7,10] and illustrated here in Equations (1)–(8). In fact, Ref. [7] tackles a similar problem and demonstrates that the general precedence formulation is the most adequate formulation for this type of problem.

The notation used in our mathematical formulation is summarized in Table 1:

**Table 1.** List of symbols for the MILP model.

| Nomenclature | |
| --- | --- |
| Indices | |
| $j,\ j'$ | Job |
| $w$ | Workstation |
| $o,\ o'$ | Operation |
| $s,\ s'$ | Stage |
| $\beta$ | Storage area |
| $b, b'$ | Single $-$ unit buffer |
| Sets | |
| $J$ | Jobs |
| $W$ | Workstations |
| $W_s$ | Workstations in stage S |
| $O_j$ | Operations of job j $\in$ J |
| $O_s$ | Operations to be assigned to stage S |
| $J^{sa}$ | Subset of jobs J that are subassemblies |
| $J^a$ | Subset of jobs J that are assemblies |
| $O^a$ | Subset of operations O that are final operations |
| Parameters | |
| $M$ | Big $-$ M constraint constant |
| $DD$ | Due dates |
| Continuous variables | |
| $MK$ | Makespan |
| $TT$ | Total tardiness |
| $ST_o$ | Start time of operation o |
| $FT_o$ | Completion time of operation o |
| $PT_o$ | Processing time of operation o |
| Binary variables | |
| $Z_{oo'}$ | 1 if operation o is processed before $o'$, otherwise 0 |
| $Y_{ow}$ | 1 if operation o is assigned to workstation w, otherwise 0 |

Based on this notation, we define the following Mixed Integer Linear Programming model:

- Minimize Makespan (MK):

$$MK \geq FT_o \quad \forall o \in O \tag{1}$$

- Minimize Total Tardiness (TT):

$$TT \geq \sum_{0=1}^{n} FT_o - DD_j \quad \forall o \in O^a \tag{2}$$

- Allocation Constraints:

$$\sum_{w \in W_s} Y_{ow} = 1 \quad \forall s \in S,\ o \in O_s \tag{3}$$

- Time Constraints:

$$FT_o \geq ST_o + PT_o \quad \forall o \in O \tag{4}$$

$$ST_{o'} \geq FT_o \quad \forall j \in J,\ (o, o') \in O_j\ /\ o' > o \tag{5}$$

- Sequencing Constraints:

$$ST_{o'} \geq FT_o - M(1 - Z_{oo'}) - M(2 - Y_{ow} - Y_{o'w}) \quad \forall \, (s, s') \in S, \, o \in O_s, \, o' \in O'_{s'}, \, w \in W_s \cap W_{s'} \tag{6}$$

$$ST_o \geq FT_{o'} - MZ_{oo'} - M(2 - Y_{ow} - Y_{o'w}) \quad \forall \, (s, s') \in S, \, o \in O_s, \, o' \in O_{s'}, \, w \in W_s \cap W_{s'} \tag{7}$$

- Assembly Constraints:

$$ST_{o'} \geq FT_o \quad \forall \, j'' \in J \, / \, j'' = j \cup j', \, j \in J^{sa}, \, j' \in J^f, \, o \in O_j, o' \in O_{j'}, \, (o, o') \in O_{j''} \tag{8}$$

- Limited Buffer Capacity Constraint:

$$ST_o = FT_{o-1} \quad \forall \, s \in S^a, \, j \in J^f, \, t \in T_j, \, t \in T_s : s > 1 \, t' \tag{9}$$

Equations (1) and (2) define the two goals of the optimization studied in our work: the minimization of the last operation to be executed, that is, the makespan (MK) and the total tardiness (TT) of the final operations of the assembly jobs. Constraint (3) ensures that each operation can only be assigned to one workstation within its respective stage. Constraint 4 sets the ending time of operation $o \in O$, while Constraint (5) establishes the precedence relationship between operations within a job according to the predefined sequence. Formulated as big-M constraints, Constraints (6) and (7) sequence any pair of operations assigned to the same workstation so that they do not overlap with each other. Constraint (8) restricts operations $o' \in O_{j'}$ to be started if and only if all operations $o \in O_j$ have been finished being $(o, o') \in O_{j''}$ and $j \in J^{sa}$, $j' \in J^f / j'' = j \cup j'$. Finally, for the model that considers limited buffer capacity, constraint (9) ensures that jobs cannot leave a workstation if the workstation downstream is blocked. For the blocking of FJSP, no storage area is added so the constraint applies directly between consecutive workstations for a given operation. It is worth noting that, when considering limited buffer capacity, this model adds unnecessary 0-time steps to jobs, since buffers are modeled as mandatory workstations with processing time 0. This makes the optimization more complex in terms of assignments, but simpler when it comes to the number of variables.

### 3.2.2. CPO Model

CP is based on computer-based syntax, and the syntax usually depends on the solver employed. Here, we represent the problem using CPO Optimizer from IBM as in [14]. It is worth noting that decision variables in CPO are a special type of variable called interval variable $x$ whose domain $dom(x)$ is a subset $\{\bot\} \cup \{[s,e] | \in \mathbb{Z}, s < e\}$. An interval variable replaces several variables of the MILP formulation: for a given operation $o$, variables $ST_o$ and $Y_{ow}$ are now contained in the interval variable $v_o^{ops}$, $o \in O_j$, $j \in J$. Ref. [25] explains the syntax of CPO and the fundamentals of interval variables and CPO constraints. For the notation and representation of the problem, we follow [14].

The notation for the CP model of the FJSP-A-LBC is given in Table 2.

Based on this notation, the following CPO model can be defined:

- Minimize Makespan (MK):

$$minimize\left(max\left(endOf\left(v_o^{ops}\right)\right)\right) \forall o \in O \tag{10}$$

- Minimize Total Tardiness (TT):

$$minimize(sum\left(end\_eval\left(f tardiness\left(v_o^{ops}\right)\right)\right)) \forall o \in O \tag{11}$$

- Allocation Constraints:

$$alternative\left(v_{o,w}^{mops}\right) \forall s \in S, \, o \in O_s, \forall w \in W_s \tag{12}$$

- Timing Constraints:

$$endBeforeStart\left(v_{o-1,j}^{ops}, v_{o,j}^{ops}\right) \forall j \in J, \, o \in O_j \, if \, o > 0 \tag{13}$$

- Sequencing Constraints:

$$noOverlap\left(v_{o,w}^{mops}, v_{o',w}^{mops}\right) \forall \ (s, s') \in S, \ o \in O_s, \ o' \in O_{s'}, \ w \in W_s \cap W_{s'} \tag{14}$$

- Assembly Constraints:

$$endBeforeStart\left(v_o^{ops}, v_{o'}^{ops}\right) \forall \ j'' \in J \ / \ j'' = j \cup j', \ j \in J^{sa}, \ j' \in J^f, \ o \in O_j, o' \in O_{j'}, \ (o, o') \in O_{j''} \tag{15}$$

- Limited Buffer Capacity Constraint:

$$endAtStart\left(v_{o-1,j}^{ops}, v_{o,j}^{ops}\right) \forall j \in J, \ o \in O_j \ if \ o > 0 \tag{16}$$

**Table 2.** List of symbols for the CPO model.

| Nomenclature | |
|---|---|
| **Parameters** | |
| $ops_{j,o}$ | List of all operations $o \in O_j, j \in J$ that are to be assigned |
| $mops_{j,o,s,w,pt}$ | List of all possible assignments of operations $o \in O_j$, $j \in J$, workstations $w \in W_s, \ s \in S$ and processing times $PT_o$ |
| **Interval Variables** | |
| $v_o^{ops}$ | Interval variable for each operation $o \in O$ contained in $ops_{j,o}$. The interval variable is defined by a start date $ST_o$, a size $PT_o$ and and end date given by $ST_o + PT_o$. |
| $v_{o,w}^{mops}, \ optional$ | Optional interval variable for each combination contained in $mops_{j,o,s,w,pt}$. The variable is declared optional to model parallel workstations. |
| **Functions** | |
| $ftardiness()$ | CpoSegmentedFunction (A piecewise linear function defined on an interval [xmin, xmax) which is partitioned into segments such that over each segment, the function is linear. |

Equation (10) establishes the goal of the optimization as the minimization the makespan while Equation (11) establishes the goal of the optimization as the minimization of the total tardiness. Constraint (12) constrains the assignment of each operation to only one workstation of the respective stage where the operation is to be assigned. Constraint (13) ensures the precedence relation between the operations of a job. Constraint (14) forces no overlap between operations executed on the same workstation. Constraint (15) adds assembly restrictions, so the operations of a job that is an assembly cannot start until all the operations of the jobs which are its subassemblies have been completed. Finally, for the CPO model of the FJSP-A with limited buffer capacity, Constraint (16) forces consecutive operations of a job to be non-stop.

*3.3. Shipbuilding Case Data and Experiments*

For the shipbuilding case, the mapping of the workshops and operations explained in Section 2 is shown in Tables 3 and 4. Table 3 displays the workstations (cells or cabins) that belong to each workshop while Table 4 maps the stages and workstations of the FJSP-A to the real problem according to the diagram from Figure 1. To ensure experiment reproducibility, in Appendix A, we provide detailed information in Tables A1 and A2, which display the operations and durations of subblocks and blocks, respectively. Furthermore, Table A3 provides a clear representation of the assembly relationships between blocks and subblocks. Finally, Figure 2 illustrates the process flow along with the stage numbers and workstations identifying the assembly stage. Buffers are also represented for limited intermediate storage capacity instances.

**Table 3.** Manufacturing cells (workstations) of each workshop.

| Workshop | Workstations (Cells) |
|----------|----------------------|
| WA1 | w1, w2, w3, w4, w5, w6, w7, w8, w9, |
| WA2 | w10, w11, w12, w13, w14, w15, w16, w17, w18 |
| WO1 | w19, w20, w21, w22 |
| WO2 | w23, w24, w25, w26 |
| BTC | w27, w28, w29, w30 |
| PC | w31, w32, w33, w34 |

**Table 4.** Mapping of the workstations, shipbuilding processes, and stages.

| Stage | Stage Name | Workstations |
|-------|------------|--------------|
| s1 | Subblock Assembly 1 | w1, w2, w3, w4, w5, w6, w7, w8, w9, w10, w11, w12, w13, w14, w15, w16, w17, w18 |
| s2 | Subblock Assembly 2 | w1, w2, w3, w4, w5, w6, w7, w8, w9, w10, w11, w12, w13, w14, w15, w16, w17, w18, w19, w20, w21, w22, w23, w24, w25, w26 |
| s3 | Turning | w27, w28, w29, w30 |
| s4 | Block Assembly | w1, w2, w3, w4, w5, w6, w7, w8, w9, w10, w11, w12, w13, w14, w15, w16, w17, w18, w19, w20, w21, w22, w23, w24, w25, w26, w27, w28, w29, w30 |
| s5 | Outfitting 1 | w1, w2, w3, w4, w5, w6, w7, w8, w9, w10, w11, w12, w13, w14, w15, w16, w17, w18, w19, w20, w21, w22, w23, w24, w25, w26 |
| s6 | Painting/Blasting | w31, w32, w33, w34 |
| s7 | Outfitting 2 | w23, w24, w25, w26 |

Different problem instances of increasing complexity have been considered for this problem. While stages remain constant, we vary the number of jobs (blocks and subblocks) and subsequent assembly operations to assess the problem's scalability. To examine the impact of limited buffer capacity, for each problem instance, we consider three scenarios: infinite or no limited buffer capacity (NBC), zero-unit buffer capacity (0BC), and single-unit buffer (1BC) capacity. In addition, for each of these scenarios, we focus on optimizing two key objectives: the makespan (MK) and the minimization of the total tardiness of all jobs (DD). Table 5 presents a summary of the experiments designed for the shipbuilding case.

**Table 5.** Instances defined for the shipbuilding case.

| Problem | Blocks × Subblocks (N × M) | Infinite Buffer Capacity | 0 Buffer Capacity | 1-Unit Buffer Capacity | MK | DD |
|---------|---------------------------|--------------------------|-------------------|------------------------|----|----|
| SB-01-NBC-MK | 5 × 10 | x | | | x | |
| SB-01-0BC-MK | 5 × 10 | | x | | x | |
| SB-01-1BC-MK | 5 × 10 | | | x | x | |
| SB-01-NBC-DD | 5 × 10 | x | | | | x |
| SB-01-0BC-DD | 5 × 10 | | x | | | x |
| SB-01-1BC-DD | 5 × 10 | | | x | | x |
| SB-02-NBC-MK | 10 × 20 | x | | | x | |
| SB-02-0BC-MK | 10 × 20 | | x | | x | |
| SB-02-1BC-MK | 10 × 20 | | | x | x | |
| SB-02-NBC-DD | 10 × 20 | x | | | | x |
| SB-02-0BC-DD | 10 × 20 | | x | | | x |
| SB-02-1BC-DD | 10 × 20 | | | x | | x |
| SB-03-NBC-MK | 25 × 50 | x | | | x | |
| SB-03-0BC-MK | 25 × 50 | | x | | x | |
| SB-03-1BC-MK | 25 × 50 | | | x | x | |
| SB-03-NBC-DD | 25 × 50 | x | | | | x |
| SB-03-0BC-DD | 25 × 50 | | x | | | x |
| SB-03-1BC-DD | 25 × 50 | | | x | | x |

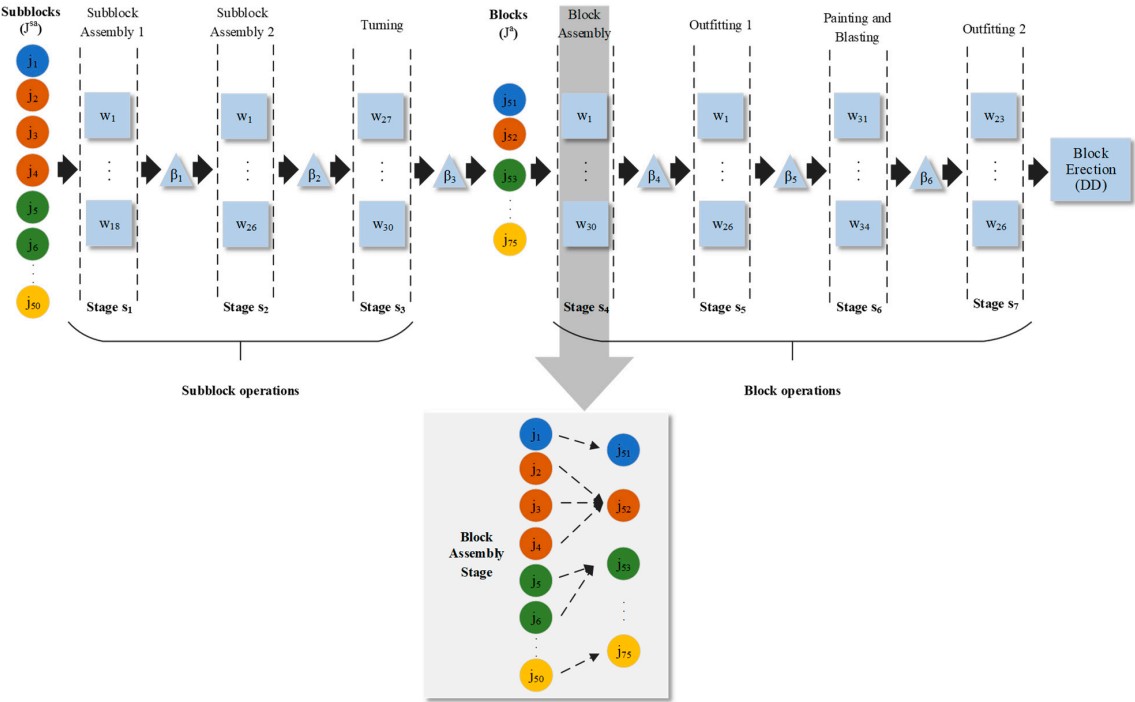

**Figure 2.** Detailed process flow of the production of subblocks and blocks.

### 3.4. Experimental Setup

The workflow designed for the present work is illustrated in Figure 3. Python 3.9 and Python APIs, provided by Gurobi (Gurobipy [41]) and CPLEX (docplex.cp [42]), respectively, were used to program the models. Gurobi Optimizer 10.0.1 was used as the optimization engine for all MILP models, while CP Optimizer 22.1.0.0 was used for all CP models. The computational experiments were conducted on a 14-core 12th Gen Intel(R) Core(TM) i7-12700H 2.70 GHz processor.

To ensure a streamlined workflow, data were automatically imported from Excel files at runtime using the Pandas library. The user defines the case number and the main parameters such as time limit and search strategy beforehand. Once the optimization is run, calls are made to the optimizer that returns the solution once the time limit is reached or the optimality gap is reduced to 0%. Output data consisting of the list of jobs' start and end times and workstations' assignments are automatically exported to an Excel file by using the Matplotlib library. The first Gantt diagram is also built using this library for a first check.

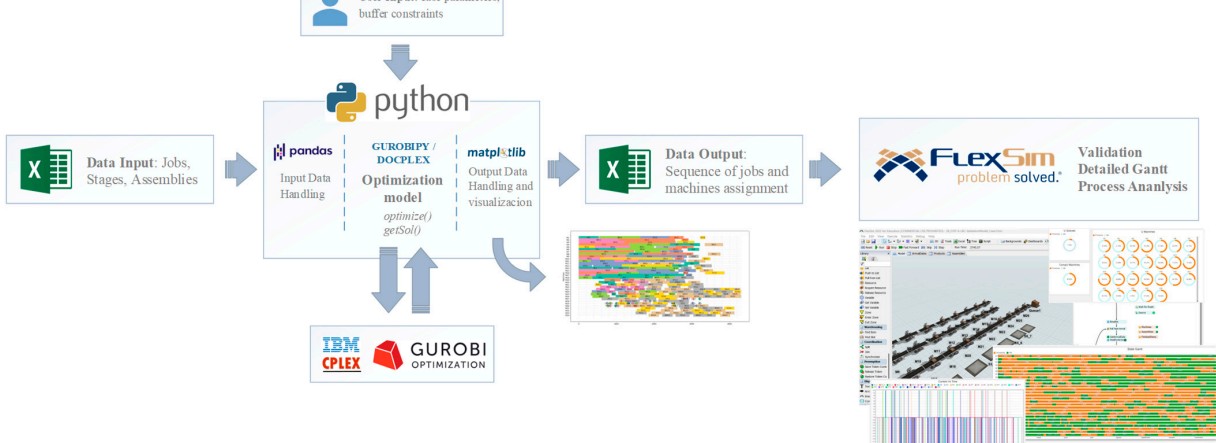

**Figure 3.** Workflow designed for the experimentation phase.

To evaluate the validity and robustness of the solution, FlexSim 22.0.8 was used to create the discrete-event simulation models. The output data from the optimization were seamlessly imported into the corresponding simulation model by using the Import/Export Excel module provided by FlexSim. The simulation model also allows for a comprehensive performance analysis of the solution, including the utilization of workstations and buffers. Additionally, during the model run, a more detailed Gantt chart is automatically generated, facilitating in-depth analysis at various simulation points. The model continuously verifies that job completion times and assembly requirements are consistently met throughout the simulation, promptly alerting the user in the event of any inconsistency.

Importantly, the model is designed to accommodate diverse types of buffer constraints, requiring only a single model for each specific case. It is worth noting that data transfer processes are fully automated, eliminating the need for time-consuming manual data management tasks.

## 4. Results and Discussion

In this section, we implemented the MILP and CP models to compare their performance and scalability. The evaluation is based on the small example and three real-world case studies previously studied by [10]. The aim is to determine the suitability of using CPO as a scheduling optimization approach for industrial-size cases. While [10] published the results of the MILP models, we have updated the results in this paper to account for computational characteristics and software adjustments.

As a second step, we applied the MILP and CP models to the FJSP-A-LBC instances presented in Table 5 based on the shipbuilding industry. In each case, we compared the computational efficiency of the two approaches and evaluated the impact of limited buffer capacity for two different objectives: the minimization of the makespan and the minimization of the total tardiness. To validate the results, we employed a discrete-event simulation model, which allowed us to draw specific conclusions regarding the scheduling outcomes.

For the cases studied in [10], we followed the termination criterion for MILP optimizations based on either a 0% integrality gap or a maximum CPU time of 3600 s. For the cases of FSJP-A with limited buffer capacity, a termination criterion of either a 0% integrality gap or a maximum CPU time of 300 s (5 min) was considered. It is worth noting that in real-world applications, shorter CPU times are typically required for prompt decision making. Therefore, we consider CPU times longer than 5 min as impractical for the actual implementation of optimization in real-world case scenarios.

### 4.1. Case Studies

The case studies examined in this research comprise the small illustrative example (CS0) and three distinct case studies (CS1, CS2, CS3) selected from Sections 5.1, 5.2, 5.3, and 5.4 of [10]. Of specific interest, CS2 has been subdivided into three progressive complexity levels: CS2.1, CS2.2, and CS2.3, corresponding to 4, 8, and 12 molds. For a comprehensive understanding of the case studies' intricacies and complexities, we refer the readers to the original paper.

In the interest of brevity, we present the optimized results in Table 6 without extensive elaboration. The CPO search strategy has been kept as default since no alternative search strategy has been proven as more effective for the problems addressed. The CPO automatic search is based on failure-directed search and iterative diving [26,43].

Given its simplicity in terms of the number of jobs and stages, CS0 serves to demonstrate the convergence of the constrained programming model to the optimal solution of the problem. Moving on to CS1, it represents an industrial-size instance of the FJSP without assemblies. It is remarkable how the CPO model is capable of reaching the optimal solution in only 2 s, compared to the iterative algorithm presented by [10], which required 1762 s and already represented a substantial time reduction over the monolithic approach. Hence, the CPO model proves highly suitable for scheduling problems of this kind without assemblies.

**Table 6.** Results.

| Problem | | | | MILP | | | MILP [10] | | CP | | |
|---|---|---|---|---|---|---|---|---|---|---|---|
| Name | J.* | S.* | A.* | Obj. | GAP | CPU Time (s) | Obj. | CPU Time (s) | Obj.* | GAP | CPU Time (s) |
| CS0 | 9 | 3 | x | 31 | 0.00% | 0.25 | 31 | 0.98 | 31 | 0.00% | 0.02 |
| CS1 | 79 | 24 | | 24,774.2 | 48.86% | 3600 | 23,015 | 1762 | 22,930.1 | 0.00% | 1.26 |
| CS2.1 (4m) | 24 | 9 | x | 979 | 0.00% | 15.26 | 979 | 28.8 | 979 | 0.00% | 0.63 |
| CS2.2 (6m) | 36 | 9 | x | 1355 | 8.34% | 3600 | 1355 | 455.7 | 1355 | 0.00% | 0.1 |
| CS2.3 (8m) | 48 | 9 | x | 1764 | 28.04% | 3600 | 1764 | 1145 | 1764 | 0.00% | 0.25 |
| CS3 | 75 | 7 | x | 200.5 | 31.62% | 3600 | 229.6 | 1100 | 179.9 | 16.9% | 3600 |

* J., number of jobs; S., number of stages; A., assemblies; Obj., objective.

When it comes to instances of FJSP-A, the increasing complexity of CS2 illustrates how CPO is designed to tackle large-sized problems. Even in the most complex case (CS2.3), the CPO model attains the optimal solution in less than a second while the iterative algorithm needed at least 1100 s to reach the same solution. The monolithic approach fails to close the gap, leaving it at 28%.

Lastly, the case study presented by [10] based on shipbuilding (CS3) involved 75 jobs and seven stages, meaning 9275 binary variables and 226 continuous variables. Figure 4 shows the Gantt chart generated by the CPO model yielding a makespan of 179.9 days and a GAP of 16.9%. It is worth noting that this result is obtained within 50 s of optimization, and no further improvement is observed within the given optimization time. Comparatively, the CPO model is capable of achieving a solution that is 50 days shorter in terms of workdays than the solution achieved by [10] after an hour of computational time. Furthermore, Ref. [10] was able to find a solution of 202.0 days after 50 h of optimization. This particular solution is not considered in our study, as our aim is to obtain high-quality solutions within reasonable computational times that are applicable in the industry.

Based on these outcomes, we can conclude that the CP formulation is highly suitable for addressing FJSP-A, even in industrial-sized scenarios such as the one presented in this paper. Consequently, the remaining sections of the paper compare the results obtained from the MILP monolithic approach and the CP model for various problem sizes and buffer capacities.

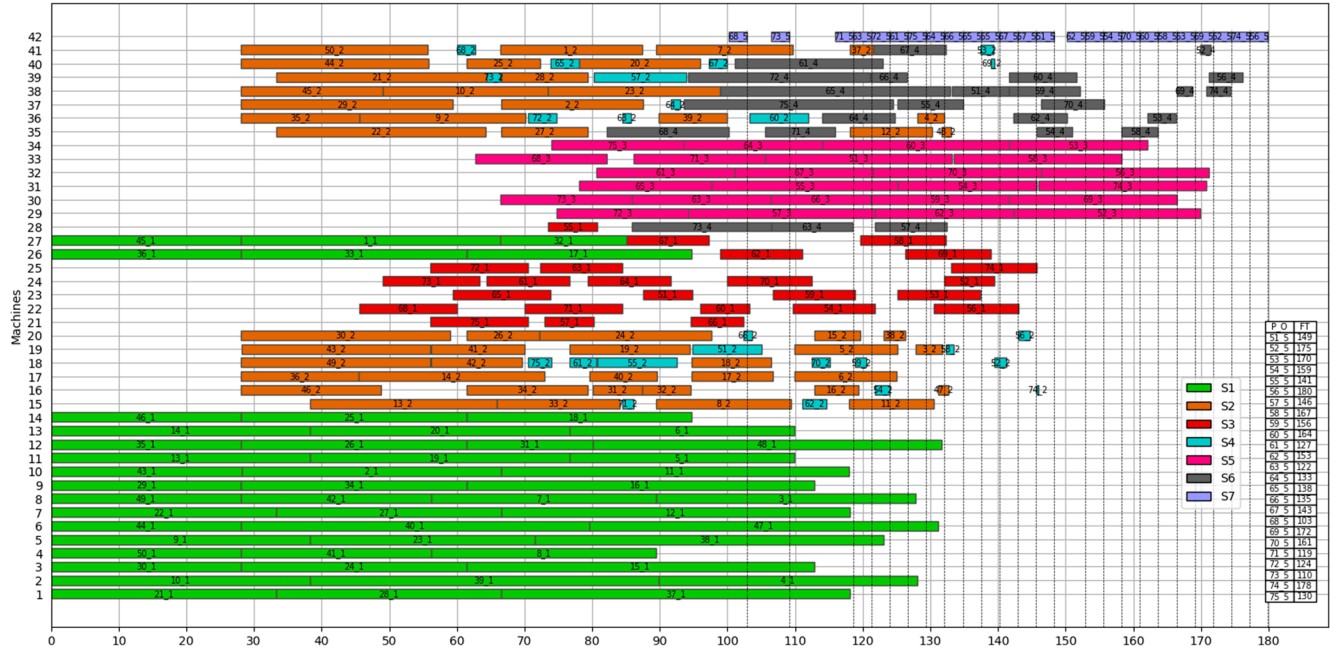

**Figure 4.** Results of the CPO model for Case Study 3 from [10].

*4.2. Shipbuilding Case*

Regarding the shipbuilding case, Table 7 presents the number of variables for the MILP and the CP model. In the MILP model, variables are shown after applying the presolve function, which transforms the problem into a smaller and more manageable equivalent form. However, it is observed that the difference in the total number of variables of both models increases as the problem size grows. For instance, considering scenario 3, the MILP model consists of up to 233 continuous variables and 19,499 binary variables, whereas the CP model comprises 4194 interval variables. Although interval variables contain more information than binary variables, it suggests that the initial size of the problem to be solved is smaller in the CP model.

**Table 7.** Number of variables and constraints in the MILP and CPO models for the shipbuilding case.

| Problem | MILP Model | | CP Model | |
|---|---|---|---|---|
| Name | Cont. Variables | Bin. Variables | Variables | Constraints |
| SB-01-NBC-MK | 47 | 1384 | 864 | 215 |
| SB-01-0BC-MK | 32 | 1369 | 864 | 246 |
| SB-01-1BC-MK | 63 | 1474 | 931 | 534 |
| SB-01-NBC-DD | 47 | 1384 | 864 | 215 |
| SB-01-0BC-DD | 34 | 1369 | 864 | 246 |
| SB-01-1BC-DD | 65 | 1474 | 931 | 534 |
| SB-02-NBC-MK | 93 | 4014 | 1694 | 396 |
| SB-02-0BC-MK | 63 | 3984 | 1694 | 458 |
| SB-02-1BC-MK | 125 | 4405 | 1823 | 1029 |
| SB-02-NBC-DD | 100 | 4014 | 1694 | 396 |
| SB-02-0BC-DD | 70 | 3984 | 1694 | 458 |
| SB-02-1BC-DD | 132 | 4405 | 1823 | 1029 |
| SB-03-NBC-MK | 233 | 19,499 | 4194 | 951 |
| SB-03-0BC-MK | 158 | 19,424 | 4194 | 1108 |
| SB-03-1BC-MK | 315 | 22,120 | 4513 | 2554 |
| SB-03-NBC-DD | 255 | 19,499 | 4194 | 951 |
| SB-03-0BC-DD | 180 | 19,424 | 4194 | 1108 |
| SB-03-1BC-DD | 337 | 22,120 | 4513 | 2554 |

Regarding the zero-unit buffer capacity problems, the number of variables is slightly reduced in the MILP model after the presolve operation as the problem becomes more constrained. In contrast, in the CP model, the number of interval variables remains the same and there is an increase in the number of constraints.

Furthermore, the results indicate that considering buffers as machines in the current problem formulation leads to a greater increase in problem size for the MILP model. For example, in scenario 3 for MK, the MILP formulation shows an increase of 82 continuous variables and 2621 variables when transitioning from the NBC to the 1BC case, resulting in a total increase of 13.7% in the number of variables. On the other hand, the CP formulation demonstrates a smaller increase of 7.6%, with the number of interval variables rising from 4194 to 4513.

It should be noted that a direct comparison of the number of variables suggests that the CP model is more efficient in handling the problem. However, it is essential to refer to the optimization results due to the different nature of variables in each model.

Table 8 presents the optimization results of the MILP and CP models for all instances, with the objective of minimizing the makespan. The results reveal that for scenarios 1 and 2, both models are capable of closing the gap and reaching the optimal solution in less than 10 s, with the CPO model achieving virtually instant results. These findings serve to validate the formulation of the CP model, as the MILP model produces the same optimal values.

**Table 8.** Results of the MILP and CPO models for the shipbuilding case and the minimization of the makespan.

| Problem | MILP Model | | | CP | | |
|---|---|---|---|---|---|---|
| Name | MK (days) | GAP (%) | CPU Time (s) | MK (days) | GAP (%) | CPU Time (s) |
| SB-01-NBC-MK | 191 | 0.00% | 0.18 | 191 | 0.00% | 0.05 |
| SB-01-0BC-MK | 191 | 0.00% | 0.24 | 191 | 0.00% | 0.06 |
| SB-01-1BC-MK | 191 | 0.00% | 0.2 | 191 | 0.00% | 0.06 |
| SB-02-NBC-MK | 197 | 0.00% | 9.45 | 197 | 0.00% | 0.32 |
| SB-02-0BC-MK | 197 | 0.00% | 5.12 | 197 | 0.00% | 1.24 |
| SB-02-1BC-MK | 197 | 0.00% | 9.51 | 197 | 0.00% | 0.32 |
| SB-03-NBC-MK | NA * | - | 300 | 267 | 19.30% | 300 |
| SB-03-0BC-MK | NA * | - | 300 | 269 | 19.99% | 300 |
| SB-03-1BC-MK | NA * | - | 300 | 279 | 22.81% | 300 |

* A feasible solution was not generated within 300 CPUs.

However, in the case of the industrial-size scenario (scenario 3), the MILP model fails to provide a feasible solution for any of the instances within the optimization time. On the contrary, the CP model is able to obtain a feasible solution within 300 s, with a GAP of approximately 20%. Considering the size of the problem, this GAP value is deemed reasonable.

With regard to the impact of limited buffer capacity, it is found to have a negligible effect for simpler instances such as scenarios 1 and 2. However, the results demonstrate that as the intermediate storage capacity becomes more limited, the GAP becomes higher. This outcome was expected for the single-unit buffer capacity, where the increased number of variables contributes to the higher GAP. However, it was not as straightforward for the 0BC case, where only the constraints were increased.

Table 9 shows the results of the MILP and CPO models for minimizing tardiness. The MILP model failed to find a solution for all the instances of case 3 within the optimization time and was unable to determine the optimal value in scenario 2 with single-unit capacity buffers. Conversely, the CPO model was able to achieve the optimal value in scenarios 1 and 2, whereas for scenario 3, it struggled to obtain solutions with GAPs exceeding 60%. Nevertheless, it managed to provide medium-quality solutions within the time limit. The model shows high sensitivity to demanding due dates, resulting in higher GAPs and lower-quality solutions.

**Table 9.** Results of the MILP and CPO models for the shipbuilding case and the minimization of the total tardiness.

| Problem | MILP Model | | | | CP | | | |
|---|---|---|---|---|---|---|---|---|
| Name | MK (days) | TT (days) | GAP | CPU Time (s) | MK (days) | TT (days) | GAP | CPU Time (s) |
| SB-01-NBC-DD | 191 | 1 | 0.00% | 0.18 | 191 | 1 | 0.00% | 0.05 |
| SB-01-0BC-DD | 200 | 1 | 0.00% | 0.23 | 191 | 1 | 0.00% | 0.05 |
| SB-01-1BC-DD | 191 | 1 | 0.00% | 0.21 | 191 | 1 | 0.00% | 0.05 |
| SB-02-NBC-DD | 251 | 13 | 0.00% | 28.94 | 200 | 13 | 0.00% | 1.35 |
| SB-02-0BC-DD | 249 | 32 | 0.00% | 43.73 | 199 | 13 | 0.00% | 77.09 |
| SB-02-1BC-DD | 276 | 13 | 46.23% | 300 | 198 | 13 | 0.00% | 1.51 |
| SB-03-NBC-DD | - | NA * | - | 300 | 296 | 18 | 60.95% | 300 |
| SB-03-0BC-DD | - | NA * | - | 300 | 301 | 43 | 84.26% | 300 |
| SB-03-1BC-DD | - | NA * | - | 300 | 294 | 18 | 60.95% | 300 |

* A feasible solution was not generated within 300 CPUs.

The table also presents the makespan values for these cases. Notably, the makespan increases significantly compared to the optimal value for larger cases, with differences exceeding 30 days in scenario 3 when using the CP model. This emphasizes the need for a trade-off between the makespan and meeting deadlines.

Figures 5 and 6 depict the Gantt charts for the instances SB-03-1BC-MK and SB-03-1BC-DD, respectively, as solved by the CP model. From these Gantts, we can observe that the production of subblocks and blocks is more orderly and requires less intermediate storage for SB-03-1BC-DD. Conversely, the use of buffers is more intensive in the case of SB-03-1BC-MK, resulting in a more compacted production, particularly evident in the final stage, $O_2$. Additionally, Figure 7 shows the Gantt chart for the SB-03-0BC-MK instance, representing the best solution in terms of makespan.

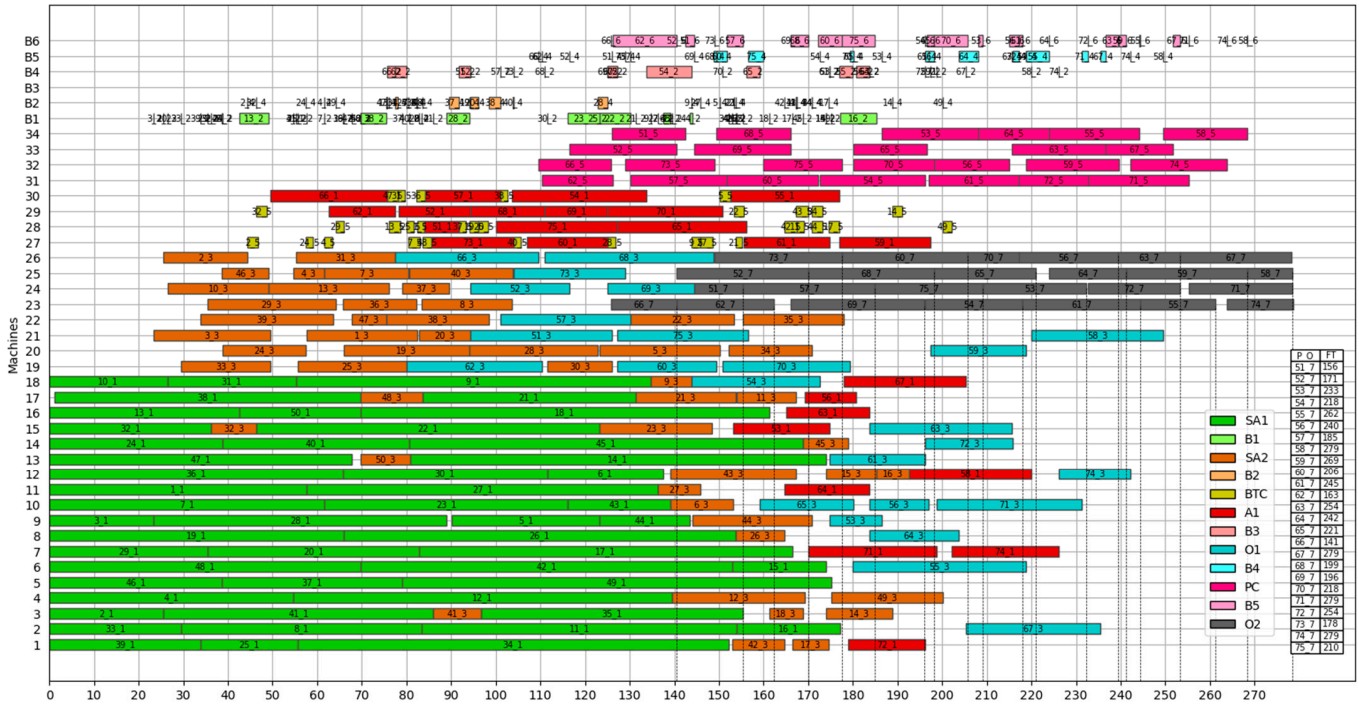

**Figure 5.** Gantt chart for the SB-03-1BC-MK instance solved by the CP model.

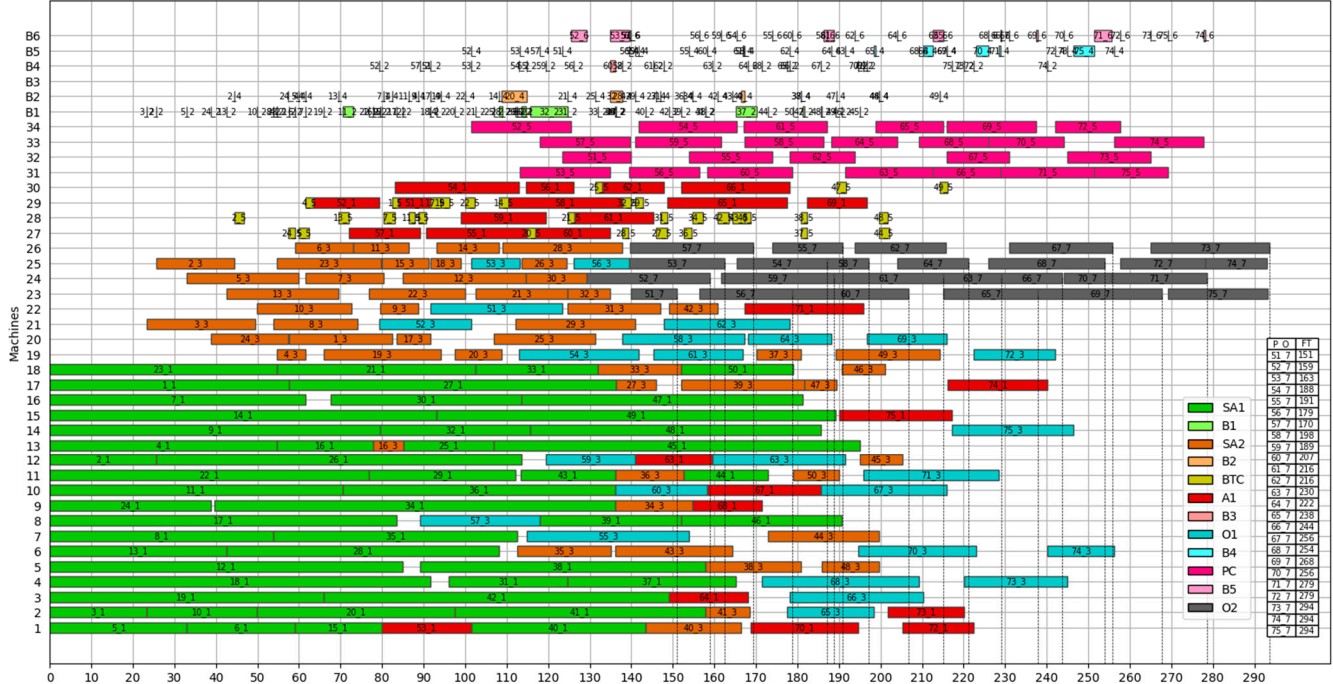

**Figure 6.** Gantt chart for the instance SB-03-1BC-DD solved by the CP model.

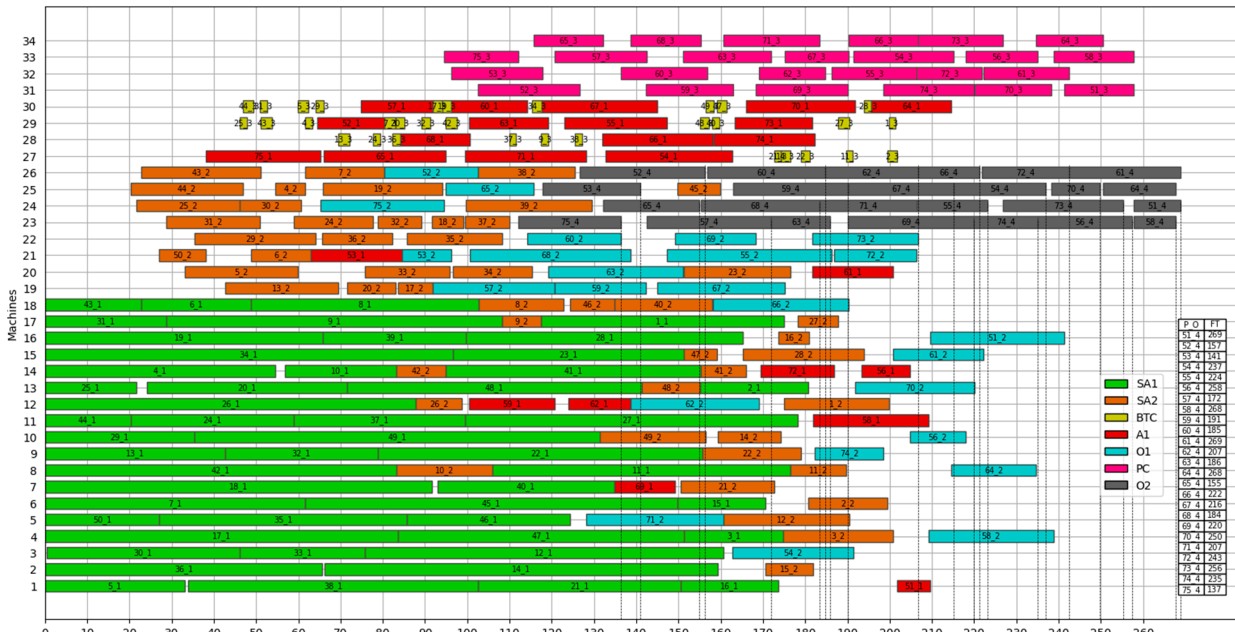

**Figure 7.** Gantt chart for the instance SB-03-NBC-MK solved by the CP model.

By utilizing the simulation model, we gained further insights into the utilization of buffers and machines. Figure 8 shows the evolution of the required intermediate storage for the SB-03-NBC-MK instance that ignores buffers. Table 10 provides details on the maximum buffer content, buffer utilization, and workstation utilization for instances based on scenario 3, as derived from the CP model.

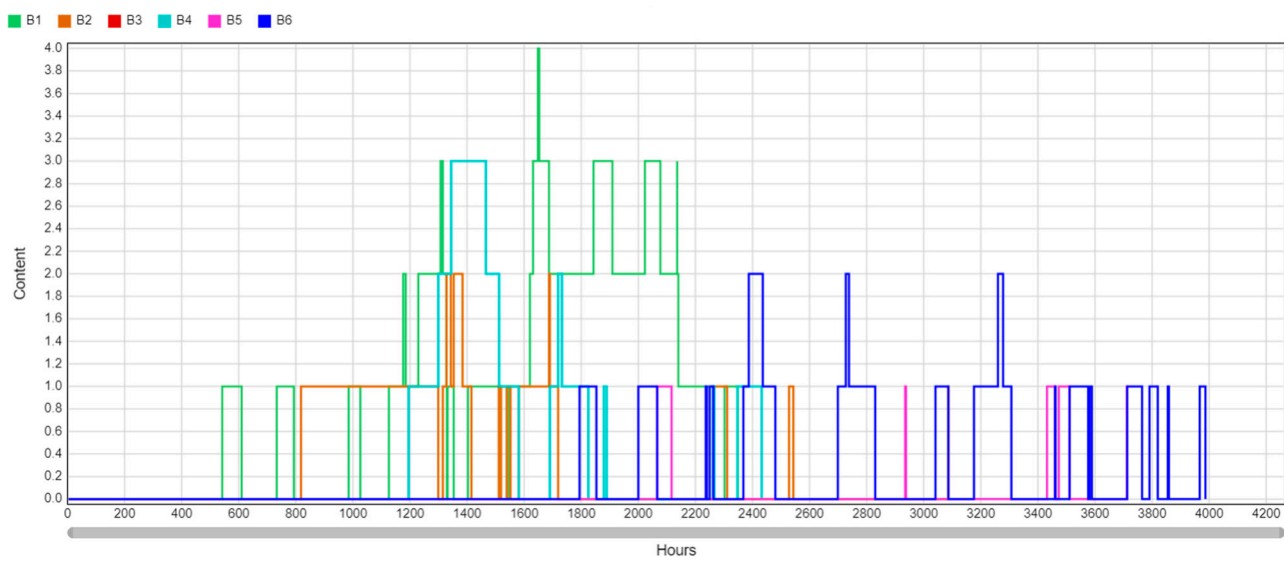

**Figure 8.** Evolution of the intermediate storage necessities for the instance SB-03-NBC-MK.

**Table 10.** Intermediate storage results for the best schedules of the shipbuilding case 3 solved by the CP model.

| CP Model | Objective | Max Content | | | | | | U Buffer | U Workstations |
|---|---|---|---|---|---|---|---|---|---|
| Case | MK (Days) | B1 | B2 | B3 | B4 | B5 | B6 | | |
| SB-03-NBC-MK | 267 | 4 | 2 | 0 | 3 | 1 | 2 | 22.73% | 71.62% |
| SB-03-0BC-MK | 269 | 0 | 0 | 0 | 0 | 0 | 0 | | 48.15% |
| SB-03-1BC-MK | 279 | 1 | 1 | 1 | 1 | 1 | 1 | 8.48% | 61.73% |

Analysis of these results reveals that the infinite-capacity buffer case exhibits a total buffer utilization of 22.73%, having several periods where up to three and two subblocks wait for the subblock assembly 2 and block turning cells, respectively, and up to three blocks wait after outfitting 1. Notably, there is even a brief period where up to four subblocks must be stored to await assembly 2 operations.

Furthermore, we observe that the utilization of cells is lower for the single-unit buffer capacity and less than 50% for the zero-unit buffer capacity case. It is our belief that accommodating subblocks and blocks in intermediate storage allows for better machine utilization, although there is no direct correlation with the makespan. As anticipated, the infinite buffer capacity case leads to the most compact schedule, albeit necessitating higher intermediate storage.

Figure 9 shows the simulation model implemented in FlexSim. Aside from analysis, this model has been instrumental in validating all instances by verifying start and finish times, as well as contents and assembly requirements. In doing so, we identified several deadlock situations (Figure 9) wherein jobs must exchange workstations simultaneously. Without intermediate buffers, these jobs obstruct one another, leading to a halt in the simulation. This is also informed by the Gantt chart generated by the simulation model where blockages are indicated in red. We believe it is crucial to consider such situations in the shipbuilding industry, particularly in terms of transport units and space management.

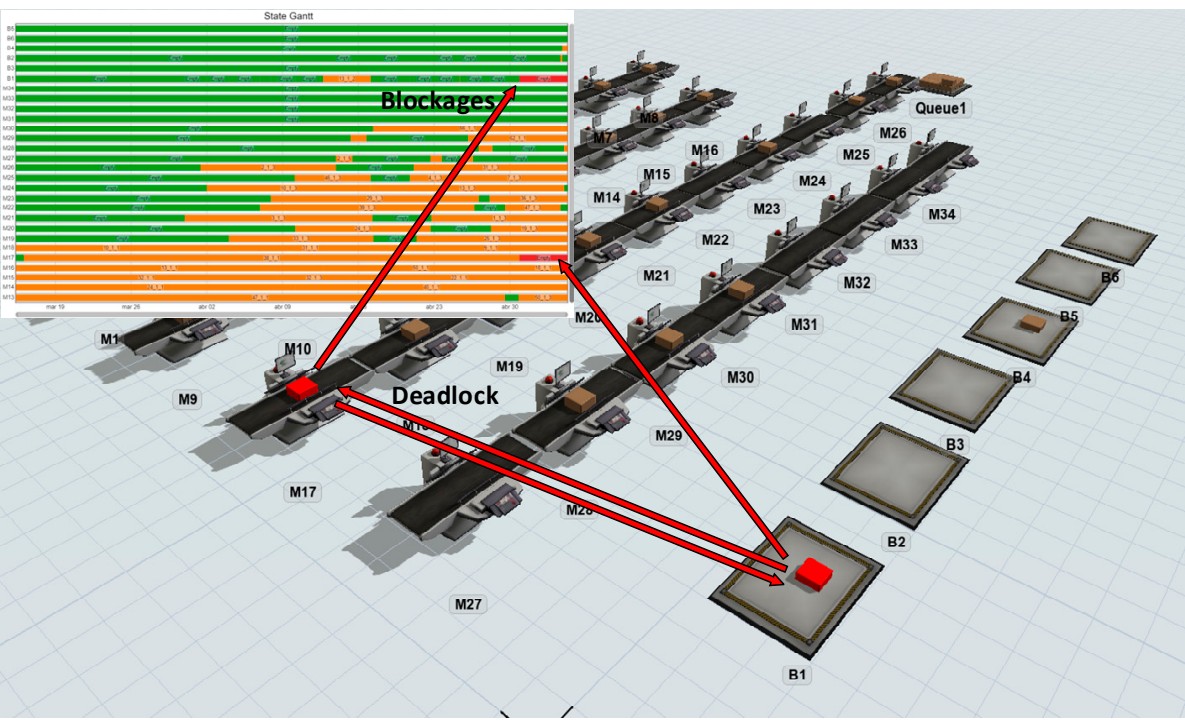

**Figure 9.** Example of deadlock produced by the schedule and detected by the DES model.

## 5. Conclusions

The present work has presented a constrained programming model for solving an industrial-size instance of the flexible job-shop problem with assemblies and limited buffer capacity. The model was initially tested on the case studies from [10], demonstrating its potential for scheduling problems involving multipurpose machines and assemblies. Following this, the model was extended to incorporate limited buffer constraints to apply it to a complex shipbuilding case and a comparison was made with a MILP model.

The results of this study confirm the validity of the proposed approach in tackling the complex problem discussed in this paper. The model consistently produced efficient solutions with reasonable gaps for the optimization of the makespan, even for industrial-scale cases involving up to 75 jobs. The inclusion of buffer constraints did not hinder

the model's ability to generate solutions, and it allowed for the evaluation of compact schedules that consider the critical spatial requirements of ship blocks and subblocks. When focusing on meeting due dates, the complexity is greatly influenced by the level of stringency imposed by these deadlines. In fact, the demanding due dates considered in this work resulted in a MILP model incapable of generating a feasible schedule and exceptionally large GAPs in the CP model. It is important to remark that the due dates play a crucial role in aligning the availability dates of blocks with the block erection necessities.

Methodologically, the combination of optimization techniques and simulation models proved valuable in assessing the solutions generated by the optimizer. It facilitated the evaluation of other key performance indexes such as machine utilization or storage requirements. One interesting insight was the appearance of potential transportation deadlock situations that could lead to shipyard logistic issues if left unaddressed (Figure 9). The simulation model also enhanced the understanding of the schedule and the quality of the solution.

Therefore, we can conclude that, overall, our approach represents a significant step towards bridging the gap between academia and the shipbuilding industry. Under conditions similar to the ones in the present study, the model demonstrates its capability to provide optimal or near-optimal solutions while considering critical aspects of the process, such as limited buffer capacity. The workflow facilitated the study of various cases in reasonable computational times, supporting our goal of providing insights and effective communication through simulation.

However, the primary limitations of the study revolve around the challenge of finding optimal solutions under highly demanding due dates or when dealing with a reduced number of machines. Moreover, we have identified an important increase in complexity when reducing the number of available machines to the extent that the CPO model is incapable of providing efficient solutions. It would be of great interest to us to explore strategies that allow reducing the number of machines per workshop while keeping acceptable makespan values. Additionally, comparing the proposed approach with other pseudo-optimal techniques such as metaheuristics could provide further insights.

Another limitation of our study is that it does not consider the block assembly strategy when optimizing the makespan, which can result in a schedule that is not aligned with the block erection strategy, potentially leading to the need for a buffer of blocks before the block erection phase. Therefore, another future research endeavor is integrating the block erection strategy in the CP model to achieve compact schedules that are aligned with the hull's construction strategy.

Furthermore, it would be of interest to integrate the optimizer and the simulation model to create a dynamic scheduler that operates under real-time conditions at different project stages. Such an integrated system would allow for real-time optimization based on recent data, allowing for minor adjustments to the scheduling to accommodate makespan objectives.

In conclusion, this work has demonstrated the effectiveness of a constrained programming model in solving complex scheduling problems in the shipbuilding industry. The combination of optimization techniques, simulation models, and the exploration of future research directions provides a solid foundation for further advancements in this field.

**Author Contributions:** Conceptualization, J.P.-Á. and D.C.-P.; methodology, J.P.-Á. and D.C.-P.; software, J.P.-Á.; validation, J.P.-Á.; formal analysis, J.P.-Á.; investigation, J.P.-Á. and D.C.-P.; data curation, J.P.-Á. and D.C.-P.; writing—original draft preparation, J.P.-Á.; writing—review and editing, J.P.-Á. and D.C.-P.; visualization, J.P.-Á.; supervision, D.C.-P. All authors have read and agreed to the published version of the manuscript.

**Funding:** This research received no external funding.

**Acknowledgments:** The authors would like to express their sincere gratitude to the authors of [10] for their invaluable support in providing the necessary data for part of the case studies conducted in this work.

**Conflicts of Interest:** The authors declare no conflict of interest.

## Appendix A

**Table A1.** List of subblocks with operations and durations for shipbuilding case 3.

| Subblock ID | Stage | Duration (h) * | Stage | Duration (h) * | Stage | Duration (h) * |
|---|---|---|---|---|---|---|
| 1 | s1 | 922 | s2 | 397 | s3 | 24 |
| 2 | s1 | 410 | s2 | 300 | s3 | 37 |
| 3 | s1 | 375 | s2 | 416 | | |
| 4 | s1 | 874 | s2 | 112 | s3 | 28 |
| 5 | s1 | 529 | s2 | 430 | s3 | 39 |
| 6 | s1 | 415 | s2 | 225 | | |
| 7 | s1 | 985 | s2 | 302 | s3 | 41 |
| 8 | s1 | 863 | s2 | 322 | | |
| 9 | s1 | 1272 | s2 | 147 | s3 | 30 |
| 10 | s1 | 424 | s2 | 363 | | |
| 11 | s1 | 1128 | s2 | 213 | s3 | 25 |
| 12 | s1 | 1359 | s2 | 476 | | |
| 13 | s1 | 682 | s2 | 429 | s3 | 42 |
| 14 | s1 | 1491 | s2 | 239 | s3 | 36 |
| 15 | s1 | 334 | s2 | 181 | | |
| 16 | s1 | 371 | s2 | 118 | | |
| 17 | s1 | 1337 | s2 | 131 | s3 | 37 |
| 18 | s1 | 1466 | s2 | 119 | | |
| 19 | s1 | 1054 | s2 | 453 | s3 | 32 |
| 20 | s1 | 760 | s2 | 183 | s3 | 33 |
| 21 | s1 | 765 | s2 | 357 | s3 | 24 |
| 22 | s1 | 1229 | s2 | 372 | s3 | 35 |
| 23 | s1 | 874 | s2 | 403 | | |
| 24 | s1 | 620 | s2 | 300 | s3 | 25 |
| 25 | s1 | 347 | s2 | 392 | s3 | 26 |
| 26 | s1 | 1406 | s2 | 174 | | |
| 27 | s1 | 1260 | s2 | 153 | s3 | 43 |
| 28 | s1 | 1048 | s2 | 460 | s3 | 26 |
| 29 | s1 | 566 | s2 | 461 | s3 | 30 |
| 30 | s1 | 733 | s2 | 233 | | |
| 31 | s1 | 459 | s2 | 356 | s3 | 26 |
| 32 | s1 | 580 | s2 | 163 | s3 | 35 |
| 33 | s1 | 473 | s2 | 320 | | |
| 34 | s1 | 1546 | s2 | 298 | s3 | 38 |
| 35 | s1 | 938 | s2 | 362 | | |
| 36 | s1 | 1051 | s2 | 264 | s3 | 31 |
| 37 | s1 | 647 | s2 | 168 | s3 | 26 |
| 38 | s1 | 1099 | s2 | 367 | s3 | 25 |
| 39 | s1 | 542 | s2 | 475 | | |
| 40 | s1 | 670 | s2 | 370 | s3 | 26 |
| 41 | s1 | 966 | s2 | 172 | | |
| 42 | s1 | 1332 | s2 | 186 | s3 | 44 |
| 43 | s1 | 366 | s2 | 452 | s3 | 44 |
| 44 | s1 | 324 | s2 | 427 | s3 | 39 |
| 45 | s1 | 1410 | s2 | 164 | | |
| 46 | s1 | 619 | s2 | 168 | | |
| 47 | s1 | 1085 | s2 | 123 | s3 | 37 |
| 48 | s1 | 1117 | s2 | 220 | s3 | 33 |
| 49 | s1 | 1536 | s2 | 400 | s3 | 32 |
| 50 | s1 | 433 | s2 | 177 | | |

* Times are provided in hours, with the consideration that 16 h is equivalent to 1 day of work.

**Table A2.** List of blocks with operations, durations, and due dates for shipbuilding case 3.

| Block ID | Stage | Duration (h) * | Stage | Duration (h) * | Stage | Duration (h) * | Stage | Duration (h) * | Due Date (h) * |
|---|---|---|---|---|---|---|---|---|---|
| 51 | s4 | 125 | s5 | 506 | s6 | 265 | s7 | 176 | 2400 |
| 52 | s4 | 256 | s5 | 355 | s6 | 383 | s7 | 475 | 2600 |
| 53 | s4 | 346 | s5 | 187 | s6 | 347 | s7 | 370 | 2700 |
| 54 | s4 | 481 | s5 | 459 | s6 | 379 | s7 | 348 | 3000 |
| 55 | s4 | 389 | s5 | 624 | s6 | 322 | s7 | 269 | 3000 |
| 56 | s4 | 184 | s5 | 212 | s6 | 271 | s7 | 358 | 3000 |
| 57 | s4 | 272 | s5 | 464 | s6 | 347 | s7 | 473 | 3100 |
| 58 | s4 | 438 | s5 | 472 | s6 | 303 | s7 | 160 | 3100 |
| 59 | s4 | 326 | s5 | 345 | s6 | 331 | s7 | 433 | 3100 |
| 60 | s4 | 289 | s5 | 355 | s6 | 325 | s7 | 448 | 3300 |
| 61 | s4 | 306 | s5 | 343 | s6 | 323 | s7 | 423 | 3300 |
| 62 | s4 | 236 | s5 | 485 | s6 | 252 | s7 | 350 | 3500 |
| 63 | s4 | 298 | s5 | 510 | s6 | 336 | s7 | 222 | 3700 |
| 64 | s4 | 304 | s5 | 320 | s6 | 254 | s7 | 276 | 3700 |
| 65 | s4 | 462 | s5 | 334 | s6 | 263 | s7 | 364 | 3900 |
| 66 | s4 | 419 | s5 | 514 | s6 | 261 | s7 | 233 | 3900 |
| 67 | s4 | 437 | s5 | 483 | s6 | 243 | s7 | 397 | 4100 |
| 68 | s4 | 265 | s5 | 607 | s6 | 267 | s7 | 448 | 4100 |
| 69 | s4 | 228 | s5 | 308 | s6 | 347 | s7 | 478 | 4300 |
| 70 | s4 | 412 | s5 | 455 | s6 | 289 | s7 | 184 | 4300 |
| 71 | s4 | 460 | s5 | 519 | s6 | 363 | s7 | 369 | 4500 |
| 72 | s4 | 276 | s5 | 314 | s6 | 248 | s7 | 329 | 4500 |
| 73 | s4 | 295 | s5 | 398 | s6 | 322 | s7 | 457 | 4700 |
| 74 | s4 | 383 | s5 | 259 | s6 | 344 | s7 | 239 | 4700 |
| 75 | s4 | 433 | s5 | 469 | s6 | 282 | s7 | 387 | 4900 |

* Times are provided in hours, with the consideration that 16 h is equivalent to 1 day of work.

**Table A3.** List of assemblies for shipbuilding case 3.

| Block ID | Subassemblies (Subblocks ID) | | |
|---|---|---|---|
| 51 | 1 | 2 | 3 |
| 52 | 4 | | |
| 53 | 5 | 6 | |
| 54 | 7 | 8 | |
| 55 | 9 | 10 | |
| 56 | 11 | 12 | |
| 57 | 13 | | |
| 58 | 14 | 15 | 16 |
| 59 | 17 | 18 | |
| 60 | 19 | 20 | |
| 61 | 21 | 22 | 23 |
| 62 | 24 | | |
| 63 | 25 | 26 | |
| 64 | 27 | 28 | |
| 65 | 29 | 30 | 31 |
| 66 | 32 | 33 | |
| 67 | 34 | 35 | |
| 68 | 36 | | |
| 69 | 37 | 38 | 39 |
| 70 | 40 | 41 | |
| 71 | 42 | 43 | |
| 72 | 44 | 45 | |
| 73 | 46 | 47 | 48 |
| 74 | 49 | | |
| 75 | 50 | | |

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
