# Peer review of "A Constrained Programming Model for the Optimization of Industrial-Scale Scheduling Problems in the Shipbuilding Industry"

_jmse, doi:10.3390/jmse11081517_

Round 1

Reviewer 1 Report

I consider that the topic is actual and scientifically interesting. The manuscript is easy to follow and the terminology is appropriate to the subject. The content of the paper is succinctly described and contextualized in relation to the presented theoretical background, also tables and figures are used effectively and support the text, also reference citations are complete and accurate.

To increase the scientific soundness, I present the following comments and recommendations:

1. I would recommend to emphasize in the abstract more strongly the relevance, originality and quality of the research, persuasively suggesting to the potential reader the items of interest that the work proposes.

2. The Section 1 Introduction is too voluminous. I recommend you condense this section by eliminating less critical information. The manuscript’s structure makes it look like a thesis chapter, which is inappropriate for a journal article. I think, in the final part of the introduction, page 4, you should emphasize more clearly the purpose of the paper and its objectives.

3. A more rigorous methodology section should be included ((perhaps by combining sections 3, 4 and 5). I recommend to present the research method much more clearly and in detail, providing the necessary elements for the reproduction of research by any other research group that uses it exactly (the repetitive and reproducible nature of science).

4. The concluding elements of the paper are represented by strong statements based on scientific arguments that are presented clearly, concisely and assumed, by argument. However, I believe that the authors should reflect the extent to which the results answered the questions mentioned in the introductory part. In my opinion, the solid argumentation of the conclusions of the paper will open new research directions and lead to the deepening of the issues studied by potential readers.

5. In Section 6 could you revise Figures 3, 5, 6, 7 so that the key dates are visible on them?

6. For the 6.1 Case studies line 421 and 6.1 Shipbuilding case line 464, the same notation is used. Correct the notation!

7. It was found that the references (number of 48) are described accurately, honestly and deontologically by the authors.

8. I recommend detailing future research opportunities that are considered to be feasible and scientifically fertile in the field.

Reviewer 2 Report

Dear Authors,

First at all, let me congratulate for the work achieved in this paper.

The strengths are related with the conceptualization, the discussions, and the literature. The use of the figures has been great. Very well done.

The weaknesses are related with the organization of the manuscript, and the information about figure and table, i.e. not all the tables or figures are explained properly.

There are a few major aspects that you should take care before its publication:

- There are some sentences difficult to follow, i.e. “[1], [12] point the fact that so far, few works have addressed the FJSP in shipbuilding. Mixed-Integer Linear Programming (MILP) [5], [9], [12] and discrete-event simulation [10], [12], [14]–[16] are the main approaches used in the area.”

- There is not too much information about the software.

Kind regards,

Reviewer 3 Report

Manuscript Number: jmse-2499179-peer-review-v1

Title: A Constrained Programming Model for the Optimization of Industrial-Scale Scheduling Problems in the Shipbuilding Industry

The aim of this manuscript is to formulate a constrained programming model for solving a flexible job-shop scheduling problem with assemblies and limited buffer capacity. The problem is based on a real case taken from the shipbuilding industry and involves the manufacturing and assembly of subblocks into blocks and the assembly of blocks for the final ship erection. The objectives considered are the minimization of the makespan and the minimization of tardiness based on ship erection due dates.

There is novelty in this paper. However, the use of English is not good. Please see my comments below.

- The MILP model in Page 8 does not follow the standard mathematical modelling structure. In particular, 1.7 and 1.8 are objective functions of the model. Consequently, they should be located before constraints 1.1 – 1.6. Likewise, correct Model 2 to have the objective function before constraints.

- The second column of Table 1 requires a heading.

- The way equations are numbered is confusing. Please number them as 1, 2, 3 instead of 1.1, 1.2, 1.3, ….

-Please proofread the manuscript. I have found some typo errors in this manuscript.

Page 2: where this papers --> where this paper

Page 2: Mixed Interger Programming --> Mixed Integer Programming

Page 3: two machine flow-shop problem --> two-machine flow-shop problem

Page 8: the sintax of --> the syntax of

Page 10: An first Gantt diagram --> The first Gantt diagram

Page 13: operations of section 2 --> operations of Section 2

Page 15: Number of variabes --> Number of variables

Please see my comments.
